

# Influence of different types of sessile epibionts on the community structure of mobile invertebrates in an eelgrass bed

Kyosuke Momota[1] and Masahiro Nakaoka[2]

[1] Graduate School of Environmental Science, Hokkaido University, Akkeshi, Hokkaido, Japan
[2] Akkeshi Marine Station, Field Science Center for Northern Biosphere, Hokkaido University, Akkeshi, Hokkaido, Japan

## ABSTRACT

Eelgrass (*Zostera marina*) beds are known to have high ecological and economical values within coastal ecosystems of the temperate northern hemisphere although their biodiversity and functions varied greatly from sites to sites. The variation in the biomass, abundance and diversity of mobile invertebrates in eelgrass beds has been examined in relation to various abiotic and biotic factors, such as water temperature, salinity, eelgrass biomass and epiphytic microalgae presence. However, the importance of sessile epibionts, such as macroalgae and calcific spirorbid polychaetes attached to eelgrass blades, has not been the focus of previous studies. In the present study, we examined the effects of three different sessile epibionts, namely, branched red algae, filamentous green algae, and calcific spirorbid polychaetes, on the biomass and diversity of mobile invertebrates in the eelgrass beds of Akkeshi in northeastern Japan. The relationships between seven abiotic and biotic variables including three types of epibionts, and biomass of 11 dominant mobile invertebrate species as well as three community-level variables (the total biomass of mobile invertebrates, species richness and the Shannon-Wiener species diversity index) were analyzed using a linear mixed model. Our results show that branched red algae are correlated with *Pontogeneia rostrata*, *Lacuna* spp., *Nereis* sp., *Syllis* sp. and the total biomass of mobile invertebrates, filamentous green algae with *P. rostrata*, *Ansola angustata* and the species diversity of mobile invertebrates, and spirorbid polychaetes with *A. angustata*, *Lacuna* spp., *Siphonacmea oblongata*, *Syllis* sp., the species richness and diversity of mobile invertebrates. The effect size of the epibionts was similar or even higher than that of abiotic and eelgrass factors on the total biomass of mobile invertebrates, species richness, species diversity and most of dominant invertebrate populations across the taxonomic groups. Consequently, epibiotic macroalgae and spirorbid polychaetes can be good predictors of the variation in the total biomass, species richness and species diversity of mobile invertebrates and the biomass of major dominant species, especially for species that have a relatively high dependency on eelgrass blades. These results suggest that the different functional groups of sessile epibionts have significant roles in determining the biomass and diversity of mobile invertebrates in eelgrass beds.

Corresponding author
Kyosuke Momota,
kyo.momota@gmail.com

## INTRODUCTION

The abundance, biomass and species diversity of marine benthic invertebrate communities vary greatly with multiple abiotic/biotic factors. The effects of temperature and salinity as environmental filters have been known to be critical factors that regulate population/ community patterns and processes in coastal habitats, especially in estuaries where strong environmental gradients are generated by tidal fluctuation and freshwater inflow (e.g., *Ysebaert et al., 2003*; *Yamada et al., 2007b*; *Douglass et al., 2010*; *Blake & Duffy, 2010*). Water temperature can either increase or decrease the abundance and diversity of component species (e.g., *Somero, 2002*; *Harley et al., 2006*; *Hoegh-Guldberg & Bruno, 2010*; *Meager, Schlacher & Green, 2011*), whereas a decrease in salinity generally leads to a lower species diversity and higher dominance by tolerant species (e.g., *Ysebaert et al., 2003*; *Yamada et al., 2007b*). Marine plants act as both a food resource because plant resource utilizers dominate in marine benthic invertebrate communities (e.g., *Valentine & Heck Jr, 1999*; *Harley, 2006*; *Poore et al., 2012*) and as habitat-former (e.g., *Attrill, Strong & Rowden, 2000*; *Lee, Fong & Wu, 2001*; *Thomsen, 2010*; *Gartner et al., 2013*).

Eelgrass (*Zostera marina*) is an important marine foundation species that is widely distributed along the coast of the northern hemisphere (*Hughes et al., 2009*). The complex physical structures created by eelgrass provide a habitat for many organisms (*Jernakoff, Brearley & Nielsen, 1996*; *Heck Jr, Hays & Orth, 2003*), which leads to an enhanced biodiversity and secondary production (*Hemminga & Duarte, 2000*; *Duffy, 2006*; *Valentine & Duffy, 2006*). A benthic invertebrate community in the above-ground parts of seagrass beds mainly consists of small crustaceans, gastropod mollusks and polychaetes, most of which are herbivores and detritivores (*Valentine & Heck Jr, 1999*; *Heck Jr et al., 2000*). These invertebrates play an important role in mediating the energy flow in the eelgrass bed ecosystem (*Duffy & Hay, 2000*; *Duffy, Richardson & France, 2005*). To explore plant-animal interactions in eelgrass bed communities, many studies have investigated the relationship between animal abundance and various eelgrass traits, such as biomass, shoot density, leaf length, habitat patch structure, and epiphytic microalgal (e.g., diatoms) biomass that serve as food resources (*Attrill, Strong & Rowden, 2000*; *Gartner et al., 2013*; *Whalen, Duffy & Grace, 2013*). However, large epibiotic organisms, such as macroalgae and sessile animals (e.g., spirorbid polychaetes, tunicates, bryozoans, hydrozoans), attached to eelgrass blades can also affect the mobile invertebrate community through resource provisioning and/or habitat modification. Despite some studies noting that the role of macroalgae on seagrass blades as a food resource or as a habitat provision can be one of the determinants of the abundance of mobile invertebrates (*Valentine & Duffy, 2006*; *Gartner et al., 2013*; *Whalen, Duffy & Grace, 2013*), most studies have focused only on the importance of seagrass and/or microalgae (e.g., *Jernakoff, Brearley & Nielsen, 1996*; *Heck Jr & Valentine, 2006*). Whilst relevant studies are few, the sessile organisms such as invasive tunicates and juvenile bay scallops attaching on eelgrass blade can affect mobile invertebrates either by providing refuge from predation (*Long & Grosholz, 2015*) or by becoming a food resource (*Lefcheck et al., 2014*). Interpreting variations in the mobile invertebrate community in relation to various functional groups of epibiotic organisms that differ in size, morphology, habitat

requirement and life history traits is thus necessary to deepen our understanding of the organization of animal assemblages in eelgrass beds and of the influences these organisms have on each other and on eelgrass.

An extensive eelgrass meadow, consisting mostly of *Zostera marina* and partly of *Z. japonica* and *Ruppia maritima*, is located in the Akkeshi-ko estuary and the Akkeshi Bay in eastern Hokkaido, Japan (*Hasegawa, Hori & Mukai, 2007*). From early summer to late fall, a large variety of algae and sessile animals (epibiotic species), which attach to eelgrass blades, are observed, including microalgae, branched red algae, *Neosiphonia* sp., *Chondria dasyphylla*, filamentous green algae, *Cladophora* sp., calcareous algae, *Pneophyllum zostericola*, and spirorbid polychaetes, such as *Neodexiospira brasiliensis*, bryozoans, hydrozoans, and tunicates. Among them, microalgae, the branched red algae and the spirorbid polychaetes are dominant in eelgrass beds for a long term, between early summer and late fall, with the peak of abundance between August and September (*Hamamoto & Mukai, 1999*; *Kasim & Mukai, 2006*; *Hasegawa, Hori & Mukai, 2007*; K Momota, 2013, unpublished data). Previous studies on benthic invertebrate assemblages in the Akkeshi-ko estuary and Akkeshi Bay have focused on their variability in relation to the salinity gradient (*Yamada et al., 2007a*; *Yamada et al., 2007b*). In addition to salinity, the spatial heterogeneity of other abiotic/biotic factors (e.g., water temperature, microalgal biomass and eelgrass biomass) is also high in estuarine systems, such as the Akkeshi-ko estuary (*Iizumi et al., 1996*; *Kasim & Mukai, 2006*; *Hasegawa, Hori & Mukai, 2007*). Nevertheless, no previous study has investigated the mobile invertebrate community structure using an approach that simultaneously accounts for the details of sessile epibionts and environmental control by abiotic factors in the seagrass beds in Akkeshi.

In the present study, we investigated how multiple abiotic and biotic factors are related to the variation in the community structure (total mobile invertebrate biomass, species richness and species diversity) of the mobile invertebrates and the population biomass of the dominant species in the eelgrass beds in Akkeshi. Our specific focus was to test the relationship between various sessile epibionts on eelgrass blades and the mobile invertebrates that live on eelgrass blades. Including these factors in the multivariate model, this analysis expands the classical models that consider only abiotic factors, eelgrass and microalgae as the explanatory variables.

## MATERIALS AND METHODS

### Study area

The Akkeshi-ko estuary (locally called Akkeshi Lake) and Akkeshi Bay are located in Northeastern Hokkaido, Japan (Fig. 1) and are connected to each other through a narrow channel (width: approximately 500 m, depth: approximately 10 m). The Akkeshi-ko is a brackish estuary, shallow water (depth range in most of the lake: 0.8–1.7 m with the maximum difference in tide levels of up to approximately ±0.6 m), with an area of approximately 32 km$^2$. Most bottom areas of the Akkeshi-ko estuary are muddy and covered with eelgrass (*Zostera marina*) except for the aquaculture farms of the clam *Venerupis philippinarum* in the intertidal zone near the channel (*Kasim & Mukai, 2006*; *Hasegawa, Hori & Mukai, 2007*;

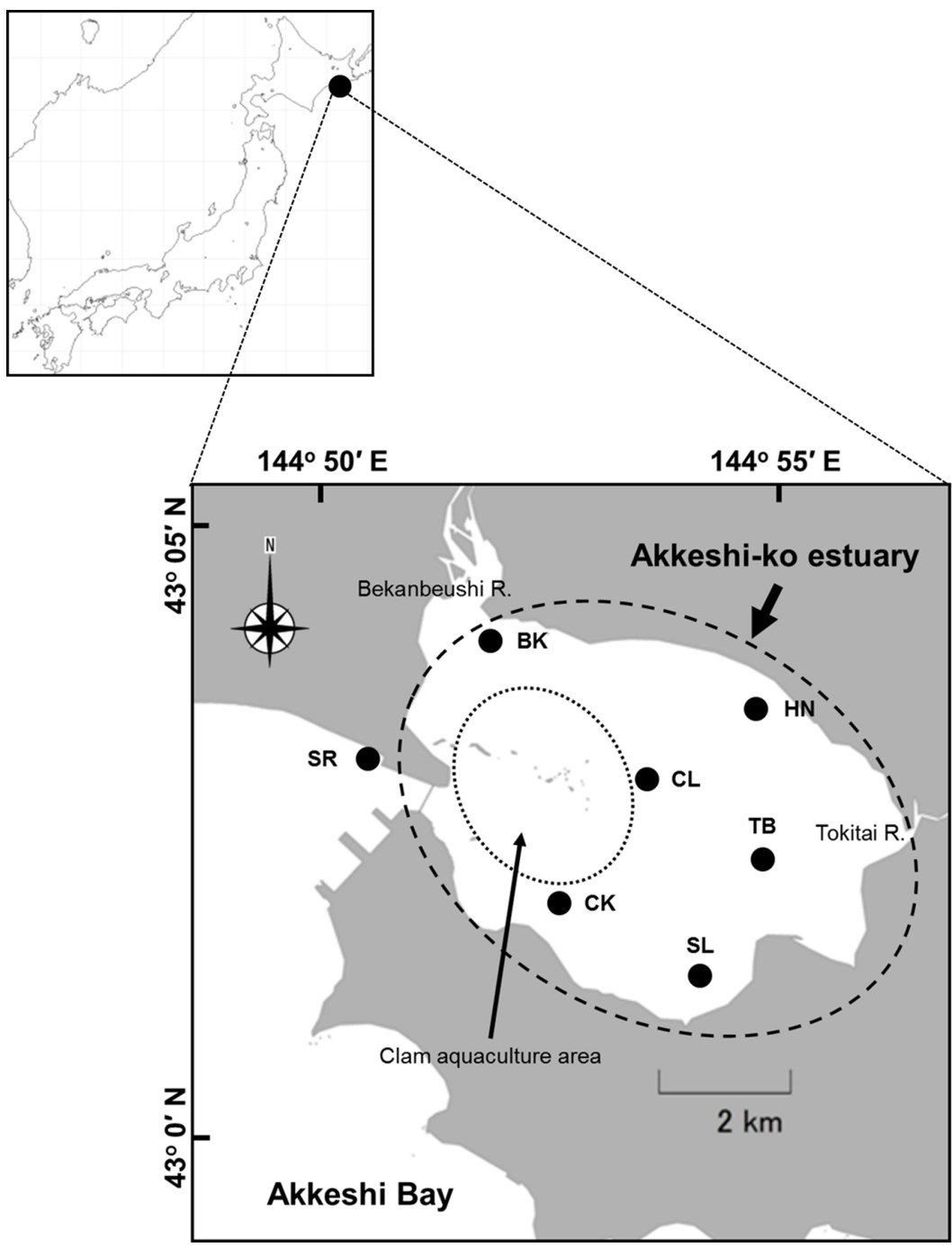

**Figure 1** **Location of the study sites in the Akkeshi-ko estuary and the Akkeshi Bay in Northeastern Japan.** The area enclosed by a dashed circle is the Akkeshi-ko estuary. Most of the clam aquaculture grounds are located in the western part of the estuary (indicated by a dotted circle).

*Yamada et al., 2007a*; *Yamada et al., 2007b*). Here, freshwater input from the Bekanbeushi River, which accounts for 98.8% of all of the flow volume (*Iizumi et al., 1996*), and tidal seawater input from the Akkeshi Bay cause steep physical and chemical environmental gradients (*Iizumi et al., 1996*; *Yamada et al., 2007a*).

Akkeshi Bay has an area of approximately 110 km$^2$ and opens to the Pacific Ocean at the south end. Two seagrass species *Z. marina* and *Z. asiatica* are present from the intertidal zone to the subtidal zone (5 m below mean low water); the former occurs at depths shallower than 2 m and the latter dominates in deeper water (*Watanabe, Nakaoka & Mukai, 2005*). The influence of the freshwater discharge on species composition of seagrass community is observed near the channel connecting the bay to the Akkeshi-ko estuary (*Yamada et al., 2007a*).

In this study, we established stations in the Akkeshi-ko estuary (BK: river mouth of the Bekanbeushi River, HN: Horonitai, TB: Toubai, SL: the southern lake, CL: the central lake and CK: Chikarakotan) and one station in Akkeshi Bay (SR: Shinryu) (Fig. 1). BK (mean sea level, MSL hereafter: 0.9 m) is located at the mouth of the Bekanbeushi River and is strongly affected by freshwater inflow. The vegetation is dense with small-sized *Z. marina* (average shoot length of 1.0 m in August). HN (MSL: 1.1 m) is in a location with a high water temperature and medium salinity relative to the other stations. In addition to *Z. marina*, *Ruppia maritima*, a seagrass species that is more tolerant to low-saline water, occurs at HN. The eelgrass beds at HN are mostly continuous but have some gaps, and the average shoot length in August is 1.3 m. TB (MSL: 1.1 m) and SL (MSL: 1.0 m) have a relatively low salinity compared to that of the other stations and are the furthest stations from the Akkeshi Bay. Although these two stations are in a similar environment, the water is often more turbid and the eelgrass bed is patchier at TB than SL. SL has a higher seagrass biomass and shoot density than TB. The average shoot length of eelgrass is approximately 1.3 m in August at both of these stations. CL (MSL: 1.4 m) and CK (MSL: 1.5 m) are deeper stations with a higher salinity and are dominated by longer eelgrass (shoot length: 1.5–3.5 m at the peak season). The eelgrass at SR (MSL: 1.5 m) in the Akkeshi Bay, has a similar shoot size to that in CL and CK. Here, the dominant seagrass species changes from *Z. marina* bed to *Z. asiatica* at a depth of approximately 2 m, as mentioned above.

According to *Yamada et al. (2007a)*, salinity varies significantly among stations but does not vary greatly among seasons. During the summer (from July to August), eelgrass biomass, microalgal biomass and mobile invertebrates reach their peak (*Hasegawa, Hori & Mukai, 2007*; *Yamada et al., 2007b*). Seasonal changes in the mobile invertebrate species richness are not clearly understood (*Yamada et al., 2007b*).

## Field sampling

We conducted a field survey in August 2012. Sample collection was performed when the tidal current was slow. We collected mobile invertebrates on eelgrass blades when the water level was deeper than the average sheath length of the eelgrass at each station (BK: 20 cm; HN, TB, SL: 30 cm; CL, CK, SR: 40 cm). Because the eelgrass at our study stations is tall (>1 m) compared to the average water depth of each station, the canopy usually reaches the water surface (except for at extremely high tides). All samplings were performed under these

conditions. We targeted mobile invertebrates but excluded some species with remarkably high mobility and low dependency on eelgrass habitat, such as mysids and decapods (*Yamada et al., 2007b*), which were not quantitatively collected by our method (see below).

We measured water temperature and salinity once at each station using a memory sensor (AAQ-175 RINKO; JFE Advantech Co. Ltd., Japan). To obtain the representative values, the sensor was carefully placed approximately 50 cm from the bottom to accurately reflect the environment inside of the seagrass meadow.

We collected three replicate samples (a total of 21 samples from all stations) of mobile invertebrates, spirorbid polychaetes and epiphytic macroalgae together with the entire above-ground parts of the eelgrass using a mesh bag (bore diameter: 20 cm, mesh size: 0.1 mm) based on the mouth area of the mesh bag (314 cm$^2$). Upon collection, we counted the number of eelgrass shoots to determine shoot density. For microalgae, five replicate samples were collected per station, using separate plastic zip bags for each eelgrass shoot, because microalgae easily fell off from eelgrass blades when collected with the mesh bag.

## Laboratory procedures

Immediately after being transported to the laboratory, the microalgae were scraped from the eelgrass blades using a glass slide; separated from other organisms such as macroalgae and spirorbid polychaetes; and then filtered using glass fiber filters (Whatman GF/F filter $\varphi$ 47 mm; Whatman International Ltd., Maidstone, UK). If other organisms were present in the microalgal samples, we carefully removed them from the filters with forceps. Other epibiotic organisms collected using mesh bags were separated from the eelgrass by scraping them off with a glass slide; these organisms were classified as red algae, green algae, spirorbid polychaetes and mobile invertebrates. To obtain dry mass, eelgrass shoots, red algae, green algae, spirorbid polychaetes and filtered microalgae were dried at 60 °C for 4 days in small aluminum foil bags, and then weighed. We counted and identified the mobile invertebrates after extraction with a sieve (500 μm) and fixation with 70% ethanol. Identification of mobile invertebrates was made to the lowest taxonomical unit possible (mostly to species) using detailed guides from the literature (Gammarid amphipod: *Nishimura, 1995*; *Carlton, 2007*, Caprella amphipod, Isopod, Copepod, Cumacea: *Nishimura, 1995*; *Carlton, 2007*, Gastropod: *Okutani, 2000*, Polychaeta: *Nishimura, 1992*; *Imajima, 1996*; *Imajima, 2001*, Turbellaria: *Nishimura, 1992*, *Carlton, 2007*, Hirunoidea: *Nishimura, 1992*) and the World Register of Marine Species online database (WoRMS: http://www.marinespecies.org).

## Statistical analysis

We used, as predictors, two abiotic factors (water temperature and salinity) and six biotic factors (eelgrass biomass (g dry weight per unit area: g DW m$^{-2}$), eelgrass shoot density (shoots m$^{-2}$), microalgal biomass (g DW m$^{-2}$), red algal biomass (g DW m$^{-2}$), green algal biomass (g DW m$^{-2}$) and spirorbid polychaete biomass (g DW m$^{-2}$)). For eelgrass biomass, we used the dry weight data collected using mesh bags. Because microalgal biomass was collected by a different sampling procedure from other biotic variables, we used the mean value of five replicates. In this study, one of our interests was the effects of morphological traits of macroalgae and spirorbid polychaetes. Thus, we separated red and

green algae by a morphological trait (red algae: branched, green algae: filamentous). All invertebrate biomass (mg ash-free dry weight per unit area: mg AFDW m$^{-2}$) was estimated from the abundance and the size fraction using the empirical equations in *Edgar (1990)*.

To test which of the eight biotic/abiotic factors was a likely predictor of the variation in the mobile invertebrate community, we fitted linear mixed models (LMMs) with a Gaussian distribution (*Bolker et al., 2009*). The station was used as a random variable. As response variables, we used the biomass of 11 dominant species for the population-level analyses, and total invertebrate biomass, species richness and species diversity (Shannon-Wiener diversity index; calculated based on biomass data) for community-level analyses. The 11 most dominant species were selected by a threshold whereby the biomass proportion accounted for more than 1% of the total invertebrate biomass (see Table S1). *Ostreobdella kakibir* (Hirudinea) was omitted from the analysis because it occurred only at one station (SR), even though they satisfied the requirement. R software (version 3.1.3) was used for all of the analyses (*R Development Core Team, 2015*).

Prior to the LMM fit, all of the variables excluding species diversity were square root transformed to improve homoscedasticity and meet the assumptions of normality of the LMMs after checking for normality with the Shapiro–Wilk test. To test for collinearity between the eight environmental variables, we calculated Pearson's correlation coefficients for all pairs. If the absolute value of the coefficient ($r$) was greater than 0.7, the level where collinearity does not affect model predictions (*Dormann et al., 2013*), we removed the relevant predictor as necessary. Because water temperature and microalgal biomass were highly correlated (Pearson's $r = -0.82$, $P < 0.01$), we removed microalgal biomass from the models. After this removal, we tested potential multicollinearity among the remaining predictors using the variance inflation factor (VIF) analysis with a cutoff of 10 (e.g., *Dormann et al., 2013*). VIF values were calculated using the *vif.mer* function developed by Frank (https://raw.githubusercontent.com/aufrank/R-hacks/master/mer-utils.R). However, all seven predictors were below the VIF value of 10 and remained. We therefore defined a reduced model with the seven predictors as the full model.

We fitted the LMMs using the *lmer* function in the lme4 package (*Bates et al., 2014*). To obtain *P*-values of the LMMs, we used the *lmerTest* package (*Kuznetsova, Brockhoff & Christensen, 2014*). We selected the optimal model comparing the candidate models on all combinations of the predictors by the Akaike information criterion as corrected for the small sample size (AIC$_c$: *Burnham & Anderson, 2002*). We obtained AIC$_c$ based on the maximum likelihood (ML) for comparisons among the LMMs because the restricted maximum likelihood (REML) is inappropriate in the case when the fixed structure is different between the candidate models (*Zuur et al., 2009*), but the parameters were estimated by REML. We used the *AIC$_c$ tab* function in the *bbmle* library (*Bolker & R Development Core Team, 2013*) to compare the AIC$_c$. After setting the optimal models, we obtained the standardized coefficients as effect sizes by re-fitting using standardized variables that were scaled by the sample standard deviation and centered by sample mean values.

Additionally, when the effect of water temperature was detected, we tested the relationship between mobile invertebrates and microalgal biomass which was omitted from the LMM because of the multi-collinearity with water temperature.

**Table 1 Environmental conditions at seven stations in the Akkeshi-ko estuary and Akkeshi Bay.** Abiotic factors in this study are indicated by **boldface**. For water temperature and salinity, we also presented data in August reported by the other studies.

| Factors | | Stations | | | | | | | Ref. |
|---|---|---|---|---|---|---|---|---|---|
| | | BK | HN | TB | SL | CL | CK | SR | |
| **Abiotic** | | | | | | | | | |
| Water temperature (°C) | | **23.8** | **26.1** | **25.9** | **25.5** | **21.0** | **22.6** | **22.5** | [a] |
| | | 21.4 | 22.4 | 22.9 | – | 22.5 | 20.0 | 18.8 | [b] |
| | | 18.1 | 20.3 | 20.3 | 21.0 | 18.5 | 17.3 | 16.6 | [c] |
| | | 21.7 | 24.1 | 23.8 | 23.9 | 21.9 | 22.7 | 18.8 | [d] |
| Salinity | | **25.0** | **26.4** | **27.0** | **27.1** | **29.2** | **26.3** | **29.9** | [a] |
| | | 16.8 | 28.1 | 28.4 | – | 29.6 | 32.0 | 28.6 | [b] |
| | | 16.1 | – | – | 23.9 | 26.0 | 26.5 | 29.6 | [e] |
| | | 26.7 | 25.0 | 13.6 | 22.4 | 27.4 | 28.4 | 29.9 | [c] |
| | | 21.2 | 23.6 | 26.0 | 26.2 | 26.8 | 26.7 | 29.9 | [d] |
| **Biotic** | | | | | | | | | |
| *Eelgrass factor* | | | | | | | | | |
| Dry mass (g m$^{-2}$) | Mean | 152.2 | 140.4 | 119.5 | 216.3 | 216.8 | 190.3 | 277.9 | [a] |
| | SD | 25.8 | 37.3 | 30.8 | 30.9 | 26.8 | 65.0 | 68.5 | |
| Shoot density (m$^{-2}$) | Mean | 233.7 | 85.3 | 74.7 | 159.0 | 85.3 | 85.3 | 96.0 | [a] |
| | SD | 18.5 | 18.5 | 18.5 | 18.5 | 0.0 | 18.5 | 18.5 | |
| *Epibiont dry mass* | | | | | | | | | |
| Microalgae (g m$^{-2}$) | Mean | 73.2 | 25.6 | 77.9 | 19.2 | 384.5 | 113.4 | 76.3 | [a] |
| | SD | 63.9 | 6.5 | 46.6 | 5.0 | 119.8 | 58.9 | 26.2 | |
| Red algae (g m$^{-2}$) | Mean | 0.1 | 9.0 | 0.0 | 4.1 | 0.0 | 4.6 | 0.0 | [a] |
| | SD | 0.1 | 6.0 | – | 2.2 | 0.0 | 7.6 | – | |
| Green algae (g m$^{-2}$) | Mean | 7.5 | 0.0 | 0.0 | 8.2 | 28.0 | 0.1 | 0.0 | [a] |
| | SD | 7.4 | – | – | 4.3 | 16.2 | 0.0 | – | |
| Spirorbid shell (g m$^{-2}$) | Mean | 53.5 | 21.8 | 6.8 | 0.0 | 0.0 | 1.9 | 944.3 | [a] |
| | SD | 28.0 | 18.7 | 7.6 | – | – | 3.2 | 190.6 | |

**Notes.**
[a] This study.
[b] *Iizumi et al. (1996)*.
[c] M Nakaoka et al. (2010, unpublished data).
[d] K Momota (2013, unpublished data).
[e] *Kasim & Mukai (2006)*.

# RESULTS

## Environmental factors

Water temperature was lower at the four stations (BK, CL, CK and SR) near the channel than at the other three stations in the inner parts of the estuary (HN, TB and SL) (Table 1). Salinity was lower at the lake-side stations (BK, HN, TB, SL and CK) that were influenced by freshwater inputs. For these stations, the inter-annual variation was also higher as shown by data collected by ourselves and other studies (Table 1).

Eelgrass biomass varied between 140 and 278 g DW m$^{-2}$ among the stations. It was lowest at TB, followed by HN and BK (Table 1). Eelgrass shoot density ranged between 85 and 234 shoot m$^{-2}$. It was highest at BK and second highest at SL. The mean densities were
not largely different among other stations. Microalgal biomass varied by more than ten-fold between the lowest station (SL) and the highest station (CL). In the latter, the microalgal biomass exceeded the biomass of the eelgrass. Macroalgae were not present at TB and SR. Branched red algae were dominated by *Neosiphonia* sp. and *Chondria dasyphylla*, and filamentous green algae were dominated by *Cladophora* sp. The mean biomass of red algae was highest at HN and that of green algae was highest at CL, though their biomasses were less than 15% of that of eelgrass. Spirorbid polychaetes were not present at SL and CL. They were highly abundant at SR where their biomass was more than three-fold greater than the eelgrass biomass.

## Mobile invertebrate community

A total of 32 mobile invertebrate species were collected in this study (Table S1). At taxonomic levels, polychaete worms made up 32.2% of the total biomass, followed by gastropods (31.3%), gammarid amphipods (23.0%), and isopods (8.8%). At the species level, a polychaete *Nereis* sp. was most dominant (24.6%), followed by gastropods *Lacuna* spp. (23.4%) and the gammarid amphipod *Ampithoe lacertosa* (18.0%). For an additional eight species including two gammarid amphipods (*Monocorophium* spp. and *Pontogeneia rostrata*), two isopods (*Cymodoce japonica* and *Paranthura japonica*), two gastropods (*Ansola angustata* and *Siphonacmea oblongata*) and two polychaetes (*Exogone naidina* and *Syllis* sp.), their proportions were less than 5% at most.

The mean value of the total mobile invertebrate biomass was the highest at CK and much lower at stations along the coastline (HN, TB and SL). Species richness was the highest at CL, followed by CK and was approximately the same level at the other stations (Fig. 2). The mean value of species diversity was the highest at CL and the lowest at SR (Fig. 2).

## Population level analyses

We found that each of the nine invertebrate populations belonging to gammarida, gastropoda and polychaeta was predicted by a different combination of environmental factors in the optimal models (Table 2). The effect size of three epibionts on dominant invertebrate species was either similar or larger than abiotic and eelgrass factors (Fig. 3). For two isopods, no environmental factor correlated with their biomass.

Water temperature was selected as the responsible factor for the variation of *A. lacertosa*, *Lacuna* spp. and all three polychaetes. Among them, only *Syllis* sp. showed a significant correlation (positive). The significant effect of the salinity gradient was detected for *A. angustata* (negative) and *S. oblongata* (positive).

For the two predictors relevant to the characteristics of the eelgrass bed, the aboveground biomass showed a significant positive relationship only with *Syllis* sp., whereas shoot density was significantly correlated with *Monocorophium* spp. (positive), *P. rostrata* (negative) and *E. naidina* (negative) (Table 2).

The biomasses of sessile epibionts (red algae, green algae and spirorbid polychaetes) on eelgrass blades were correlated with many invertebrate populations in different manners, excluding *A. lacertosa*, *Monocorophium* spp., two isopods and *E. naidina*. Red algal biomass was positively correlated with *P. rostrata*, *Lacuna* spp. and *Nereis* sp., but negatively correlated with *Syllis* sp. and tended to be negatively correlated with *A. angustata*. Green algal

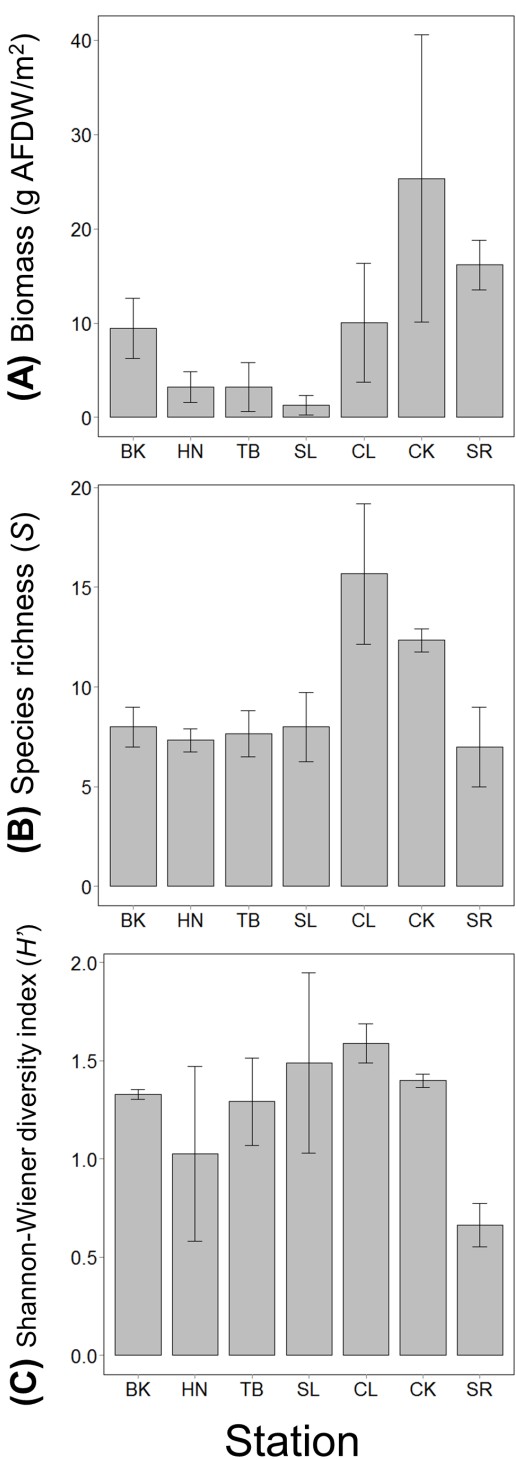

**Figure 2** **(A) The total invertebrate biomass, (B) species richness and (C) Shannon-Wiener diversity index at the seven stations in the Akkeshi-ko estuary and Akkeshi Bay.** The bars indicate the mean values with SDs. The order of the stations is lined up based on relative size of the impact of freshwater inflow or seawater from Akkeshi Bay.

**Table 2  Results of LMMs for explaining responsible environmental factors on variation in mobile invertebrate populations and community components.** $AIC_c$ scores and delta $AIC_c$ are also reported. Significant coefficients ($P$-values < 0.05 level) and the lowest $AIC_c$ scores are in **bold face**.

| Response | Model | Predictor | | | | | | | | $AIC_c$ | $\Delta AIC_c$ |
|---|---|---|---|---|---|---|---|---|---|---|---|
| | | (Intercept) | WT | Sal | ZM.bm | ZM.den | Red.alg | Grn.alg | SP.bm | | |
| **Population** | | | | | | | | | | | |
| Gammarid amphipoda | | | | | | | | | | | |
| *Ampithoe lacertosa* | Null | **33.978** | – | – | – | – | – | – | – | 193.9 | 1.0 |
| | Full | 673.282 | −87.465 | −48.498 | −0.482 | 4.427 | 7.243 | −0.761 | −0.736 | 216.7 | 23.8 |
| | Optimal | **1097.930** | −79.180 | −129.700 | | | | | | **192.9** | – |
| *Monocorophium* spp. | Null | 8.669 | – | – | – | – | – | – | – | 137.7 | 2.0 |
| | Full | 582.409 | −42.288 | −72.767 | 0.079 | 0.986 | 0.265 | 0.428 | 0.059 | 160.1 | 24.4 |
| | Optimal | −6.824 | | | | **1.469** | | | | **135.7** | – |
| *Pontogeneia rostrata* | Null | **10.068** | – | – | – | – | – | – | – | 147.2 | 19.8 |
| | Full | 247.381 | −11.576 | −32.023 | 0.407 | **−2.751** | **4.934** | **2.878** | 0.111 | 139.5 | 12.1 |
| | Optimal | **20.728** | | | | −1.842 | 5.023 | 2.523 | | **127.4** | – |
| Isopoda | | | | | | | | | | | |
| *Cymodoce japonica* | Null | **11.630** | – | – | – | – | – | – | – | **187.0** | 0.0 |
| | Full | 605.245 | −15.501 | −100.306 | 4.532 | −5.585 | 0.308 | 2.552 | −0.113 | 211.1 | 24.1 |
| | Optimal | **11.630** | | | | | | | | **187.0** | – |
| *Paranthura japonica* | Null | **14.077** | – | – | – | – | – | – | – | **173.4** | 0.0 |
| | Full | 132.132 | 8.691 | −32.148 | 0.349 | −0.924 | 2.623 | 4.411 | 0.463 | 200.0 | 26.6 |
| | Optimal | **14.077** | | | | | | | | **173.4** | – |
| Gastropoda | | | | | | | | | | | |
| *Ansola angustata* | Null | 6.014 | – | – | – | – | – | – | – | 159.6 | 10.1 |
| | Full | 555.710 | −12.507 | **−96.596** | −0.524 | 1.180 | −1.487 | 3.730 | **0.923** | 167.3 | 17.8 |
| | Optimal | **600.167** | | **−116.259** | | | −2.645 | **5.102** | **1.149** | **149.5** | – |
| *Lacuna* spp. | Null | 28.820 | – | – | – | – | – | – | – | 197.0 | 11.2 |
| | Full | 880.106 | −129.988 | −43.910 | 0.940 | −2.094 | **10.442** | −2.761 | **2.607** | 203.9 | 18.1 |
| | Optimal | 522.161 | −106.591 | | | | **10.634** | | **2.696** | **185.8** | – |
| *Siphonacmea oblongata* | Null | 8.003 | – | – | – | – | – | – | – | 166.6 | 17.2 |
| | Full | −350.288 | −16.668 | **82.450** | −1.654 | 2.567 | 1.664 | −2.471 | 1.063 | 172.7 | 23.3 |
| | Optimal | **−190.996** | | **36.374** | | | | | **1.426** | **149.4** | – |

| Response | Model | Predictor | | | | | | | | AIC$_c$ | ΔAIC$_c$ |
|---|---|---|---|---|---|---|---|---|---|---|---|
| | | (Intercept) | WT | Sal | ZM.bm | ZM.den | Red.alg | Grn.alg | SP.bm | | |
| **Polychaeta** | | | | | | | | | | | |
| *Exogone naidina* | Null | 8.274 | – | – | – | – | – | – | – | 182.1 | >0.1 |
| | Full | 1003.724 | −66.607 | −119.185 | 1.236 | −5.958 | −1.510 | 0.675 | −0.184 | 203.4 | 21.3 |
| | Optimal | **988.365** | −75.725 | −106.916 | | −4.922 | | | | **182.1** | – |
| *Nereis* sp. | Null | 23.110 | – | – | – | – | – | – | – | 211.7 | 6.4 |
| | Full | 1994.677 | −171.788 | −216.463 | 5.651 | −9.760 | **13.928** | 7.192 | 0.017 | 221.1 | 15.8 |
| | Optimal | 844.824 | −171.482 | | | | **16.967** | | | **205.3** | – |
| *Syllis* sp. | Null | **6.678** | – | – | – | – | – | – | – | 175.2 | 1.3 |
| | Full | −342.880 | **45.302** | 14.699 | **5.108** | −0.141 | **-6.889** | −1.247 | −1.140 | 191.5 | 17.6 |
| | Optimal | **−269.866** | **45.615** | | **4.886** | | **−6.616** | | **−0.908** | **173.9** | – |
| **Community component** | | | | | | | | | | | |
| Total invertebrate biomass | Null | **2.785** | – | – | – | – | – | – | – | 72.0 | 10.3 |
| | Full | 60.096 | −5.936 | −5.985 | **0.214** | −0.097 | **0.456** | 0.184 | 0.056 | 81.0 | 19.3 |
| | Optimal | 23.569 | −4.937 | | **0.219** | | **0.401** | | | **61.7** | – |
| Species richness | Null | **3.027** | – | – | – | – | – | – | – | 28.0 | 14.7 |
| | Full | **16.581** | **−1.485** | −1.211 | 0.080 | −0.094 | −0.036 | 0.080 | −0.023 | 33.6 | 20.3 |
| | Optimal | **13.909** | **−2.185** | | | | | | **−0.031** | **13.3** | – |
| Species diversity | Null | **1.255** | – | – | – | – | – | – | – | 19.9 | 8.6 |
| | Full | 2.056 | −0.126 | −0.049 | 0.010 | 0.012 | −0.068 | 0.034 | −0.026 | 36.7 | 25.4 |
| | Optimal | **1.288** | | | | | | **0.065** | **−0.020** | **11.3** | – |

**Notes.**

WT, water temperature; Sal, salinity; ZM.bm, eelgrass biomass; ZM.den, eelgrass shoot density; Red.alg, red algal biomass; Grn.alg, green algal biomass; SP.bm, spirorbid polychaete biomass.

biomasses were positively correlated with *P. rostrata* and *Lacuna* spp. The biomass of spirorbid polychaetes was positively correlated with all three species of gastropods and was negatively correlated with *Syllis* sp.

Although epiphytic microalgae were removed from our analysis because of the collinearity with water temperature, no significant correlation was found for species that were correlated with water temperature (*A. lacertosa*: Pearson's $r = 4.05$, $P = 0.25$; *Lacuna* spp.: $r = −3.42$, $P = 0.69$; *E. naidina*: $r = 0.31$, $P = 0.92$; *Nereis* sp.: $r = 10.87$, $P = 0.34$; *Syllis* sp.: $r = 0.21$, $P = 0.86$).

## Community level analyses

The total invertebrate biomass tended to decrease with water temperature, and significantly increased with increasing eelgrass biomass and red algal biomass (Table 2). Species

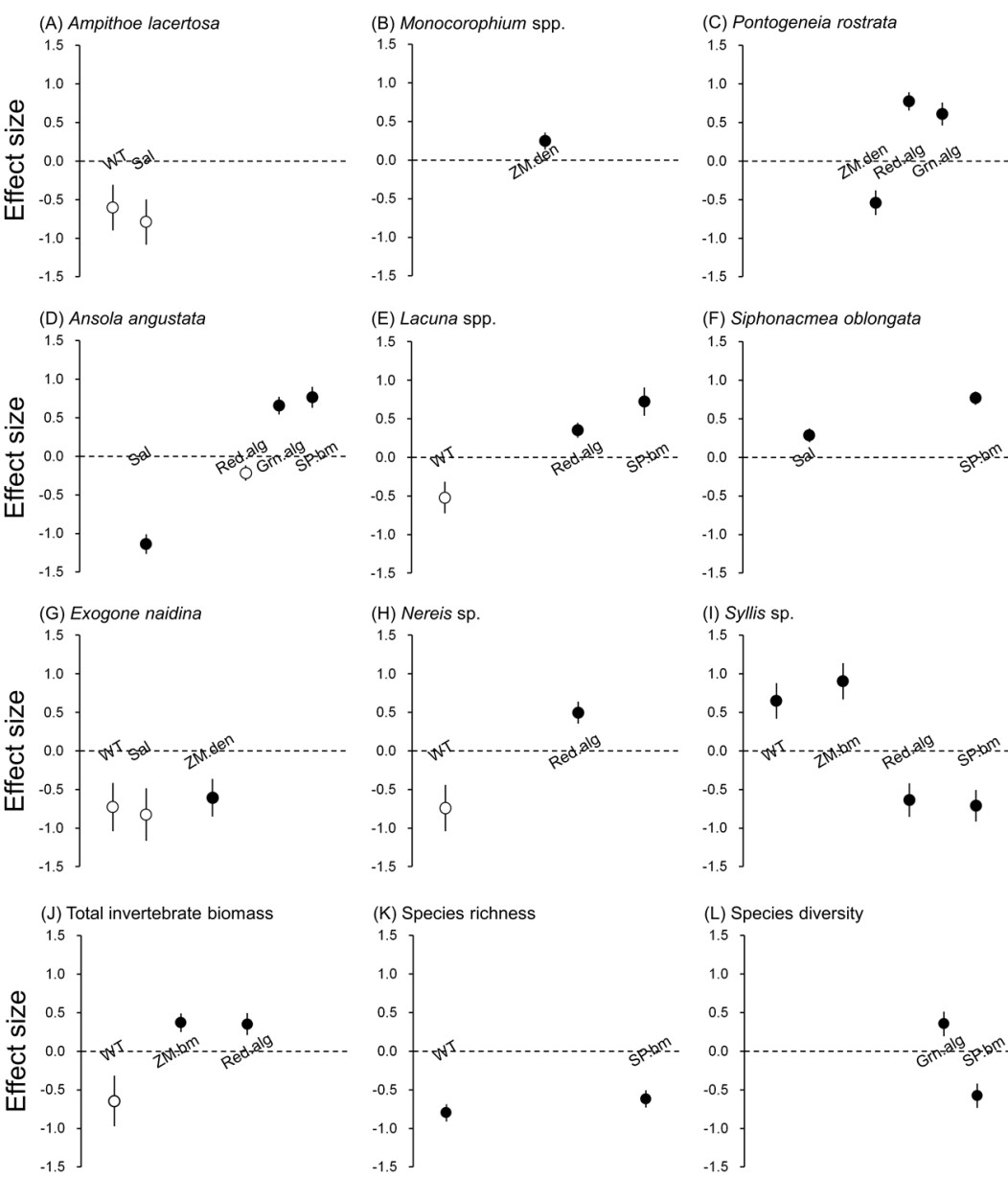

**Figure 3  Effect size of abiotic and biotic factors on mobile invertebrate populations and community detected by linear mixed models.** Water temperature (WT), salinity (Sal), eelgrass biomass (ZM.bm), eelgrass shoot density (ZM.den), branched red algae (Red.alg), filamentous green algae (Grn.alg) and spirorbid polychaetes (SP.bm) We only reported the results of predictors selected by the best models (Table 2). Open circles represent detected predictors without significance ($P > 0.05$) and filled circles represent detected predictors with significance ($P < 0.05$). Error bars indicate standard errors of effect sizes.

richness showed a negative correlation with water temperature and spirorbid polychaetes. Species diversity was positively correlated with green algal biomass, but was negatively correlated with spirorbid polychaetes (Table 2). The effect size of red algal biomass on the total invertebrate biomass was similar to that of eelgrass biomass, and that of spirorbid polychaetes on species richness was also similar to that of water temperature (Fig. 3).

## DISCUSSION

The present study demonstrated that the biomass gradient of epibiotic organisms (e.g., macroalgae and spirorbid polychaetes) was a good predictor of the variation in some dominant mobile invertebrates in the eelgrass bed and the population biomass of the community parameters such as total biomass and diversity. Further, we found that the population biomasses and community components were not always influenced only by a single factor but also by multiple factors. The effect of the macroalgae is notable because these sessile epibionts have much lower biomass compared to the biomass of eelgrass and epiphytic microalgae. However, the observed relationships between these functional groups and mobile invertebrate populations varied greatly among the species.

In the optimal models, the effects of biomass of epibiotic organisms on the gammarid amphipod *P. rostrata*, all three gastropod species (*A. angustata*, *Lacuna* sp. and *S. oblongata*) and two polychaetes (*Nereis* sp. and *Syllis* sp.) were detected. For those species, the sessile epibionts were positively related to mobile invertebrate biomasses except for *Syllis* sp. and *P. rostrata*, which showed a positive correlation with both red and green algae. The algae are considered to be used as a temporal shelter (habitat) rather than as a food resource because these animals do not firmly attach to the eelgrass blades but rather drift among shoots (*Suh & Yu, 1997*; *Yamada et al., 2007b*; *Yu, Jeong & Suh, 2008*), and because they have a preference for feeding on phytoplankton and detritus (*Yu & Suh, 2011*). High predation risk for swimming amphipods with low self-defense abilities, such as *P. rostrata*, has been reported in several studies (*Sudo & Azeta, 1992*; *Beare & Moore, 1998*). In fact, gammarid amphipods are a major source of prey for blennoid fish in the eelgrass beds of Northern Japan (*Watanabe et al., 1996*; *Sawamura, 1999*; *Yamada et al., 2010*). Therefore, the complex micro-habitat created by macroalgae allows them to escape these predators.

All three gastropods increased in correlation with spirorbid polychaetes, whereas the responses to the other factors were different (Table 2). Because the gastropods adhere to flat seagrass blades, the flat (simple) structure created by seagrass blades can be better than the rough structure of spirorbid polychaetes. Therefore, competition for space (negative effect) appears to be more expected than facilitation. Although we do not have a good answer for the positive relationships, one possibility for this unexpected result is that the rough structure acts as a shelter because small-sized individuals (<3 mm) are frequent in gastropod populations during the summer season (*A. angustata*: Momota, personal observation; *Lacuna* spp.: *Kanamori, Goshima & Mukai, 2004*; *S. oblongata*: *Toyohara, Nakaoka & Tsuchida, 2001*).

Red algae are considered to positively affect *Nereis* sp. by providing habitat because polychaetes build tubes both on eelgrass blades and in red algal canopies in Akkeshi (Momota, unpublished data). The negative effect of red algae and spirorbid polychaetes on *Syllis* sp. may suggest that this mobile polychaete prefers a simple structured habitat without a complex micro-habitat created by eelgrass blades with sessile epibionts.

In addition to the effects of sessile epibionts, the significant effects of water temperature, salinity, eelgrass biomass and shoot density were detected for a majority of the dominant species, although the patterns and directions of the effects were different among them.

Surprisingly, eelgrass biomass was not correlated with most species except for *Syllis* sp., and the direction (positive/negative) of the effect of eelgrass shoot density was different among the species. The same response of syllid polychaetes was reported in previous studies (e.g., *Bone & San Martín, 2003*). For eelgrass shoot density, the result suggests that it indirectly affects mobile invertebrates through interfering with multiple physical and biological processes (e.g., water current and flux, detritus and drifting algae trapping, recruitment, and predation intensity: *Gambi, Nowell & Jumars, 1990*; *Robbins & Bell, 1994*; *Attrill, Strong & Rowden, 2000*; *Boström & Bonsdorff, 2000*; *Lee, Fong & Wu, 2001*; *Hovel et al., 2002*). Notably, the contrasting relationships of *P. rostrata* with eelgrass shoot density and macroalgae imply that the shelter effect is different depending on the spatial scale (i.e., blade scale, shoot/patch scale).

The isopods *C. japonica* and *P. japonica* were not correlated with any abiotic or biotic factors because of the low dependency on the seagrass habitat; they can utilize other numerous habitats created by both natural and artificial materials (e.g., mussel beds, oyster reefs: *Marchini et al., 2014*; *Nakamachi, Ishida & Hirohashi, 2015*; gravel, litter layer of macrophytes, *Sargassum* meadow: Momota, personal observation). Additionally, their uniform appearance throughout all of the stations indicates that they have a wide tolerance to a broad range of environmental stress, which leads to a lack of correlation with any of the abiotic factors. Additionally, the gammarid amphipod *A. lacertosa* was not significantly correlated with any factors. This species is widely distributed along the Pacific-rim coast of the northern hemisphere and utilizes a variety of plant habitats by building tubes (*Hiebert, 2015*), which may explain why it did not show any relationship with the environmental gradients.

Although the discussion on underlying drivers that generate apparent correlations (i.e., the causalities) between epibionts and mobile invertebrates is not our main focus, the indirect effects and the top-down control of mobile invertebrates should also be taken into account to interpret present findings. For example, we can give an alternative possibility for the positive relationship between gastropods and spirorbid polychaetes such that high grazing of the gastropods facilitates the recruitment of spirorbid polychaetes through the removal of the microalgal cover.

Total biomass, species richness and species diversity were differentially correlated with abiotic/biotic factors, and varied in a complex manner although processes were unclear. The optimal model of the three community variables contains one or two variables of sessile epibionts. The positive correlation of red algae with total biomass reflects that with highly dominant species, such as *Lacuna* spp. and *Nereis* sp., which occupied more than 48% of the total biomass. The negative interaction of spirorbid polychaetes with species richness and diversity suggest that spirorbid polychaetes can decrease the homogeneity of the biomasses of component species within a community by allowing some competitive species to dominate. The effect of green algae was found only on species evenness, but not on total biomass nor on species richness, suggesting that the green algae may increase species evenness by decreasing abundance of dominant species though the actual mechanisms remain to be cleared.
## CONCLUSIONS

The present study suggests that macroalgae and sessile animals on eelgrass blades can affect the biomass and diversity of mobile invertebrates and that incorporating these biotic factors can improve the prediction of the variability of the mobile invertebrate community in the eelgrass bed. However, the underlying causal relationships appear to be complex and vary greatly from species to species. Our findings were based on data collected over one sampling period when the eelgrass bed was most productive and when the abundance and/or diversity of algae and mobile invertebrates typically reached their maximum. A more comprehensive investigation of the functional relationships among the various types of organisms and of the temporal changes should be conducted in future studies on eelgrass bed communities.

Recent studies demonstrated that the capacity for resistance and resilience to environmental stress and perturbations vary with food web structure in seagrass beds, which knowledge can contribute to improvement of coastal management (*Unsworth et al., 2015*; *Maxwell et al., 2016*; *Östman et al., 2016*). Our study comparing population and community level responses of epifauna to different types of epibionts on eelgrass blades adds more knowledge on the complex trophic/non-trophic interactions of eelgrass communities, and will promote more understandings of the resilience and the feedback mechanisms of seagrass ecosystems which offer variable ecosystem services to human such as seafood provisioning and water quality controls (*Cullen-Unsworth et al., 2014*; *Nordlund et al., 2016*).

## ACKNOWLEDGEMENTS

We wish to thank S Hamano, H Katsuragawa and other members in Akkeshi Marine Station, NM Kollars in UC Davis, Dr. K Abe and other staff in National Research Institute of Fisheries and Environment of Inland Sea, and T Maezawa in Hokkaido University.

### Funding

This study is partly supported by JST/CREST (Establishment of core technology for the preservation and regeneration of marine biodiversity and ecosystems), by JSPS/KAKENHI (no. 21241055), and by the Environment Research and Technology Development Fund (S-15 Predicting and Assessing Natural Capital and Ecosystem Services (PANCES)) of the Ministry of the Environment, Japan. The funders had no role in study design, data collection and analysis, decision to publish, or preparation of the manuscript.

### Grant Disclosures

The following grant information was disclosed by the authors:
JST/CREST.
JSPS/KAKENHI: 21241055.
Environment Research and Technology Development Fund.

### Competing Interests

The authors declare there are no competing interests.

## Author Contributions

- Kyosuke Momota conceived and designed the experiments, performed the experiments, analyzed the data, contributed reagents/materials/analysis tools, wrote the paper, prepared figures and/or tables.
- Masahiro Nakaoka conceived and designed the experiments, analyzed the data, contributed reagents/materials/analysis tools, wrote the paper.

## Data Availability

The raw data has been supplied as Data S1.

## Supplemental Information

Supplemental information for this article can be found online at http://dx.doi.org/10.7717/peerj.2952#supplemental-information.

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
