# Peer review of "Influence of different types of sessile epibionts on the community structure of mobile invertebrates in an eelgrass bed"

_PeerJ, doi:10.7717/peerj.2952_

## Round 0.1 · original submission · Minor Revisions

Both reviewers suggest that this manuscript merits minor revisions and they have made some excellent suggestions to improve it.

93 State what the relationship is between eelgrass and tunicates/scallops
229 and 252 "fitted" rather than "fit"
295 two gammarid amphipods
374 providing rather than provisioning?
416 A little explanation here of how you think green algae may impact on species eveness?

·

Basic reporting

The standard of reporting is in the main, fine - however there are some areas that might need rewording to improve clarity and these are marked on the ms. Some of the discussion is a little long and could do with shortening, especially when not directly related to the main findings. The structure is appropriate and conforms to the journal templates. There is good use of the literature and the study does add to the main body of work surrounding the subject area.

Experimental design

There are some clear research questions, but in some places these are not very clearly worded or the sampling regime surrounding this detailed effectively. One concern is that only three 'replicates' were analysed at each station and that some of the fauna on the eelgrass was not sampled as it did not directly use that environment or the epiphytes on the seagrass. If the authors can show that this is typical of research in the field (in terms of sample size) then that is fine, but it is a little light on replication. Whilst crustaceans might not feed on the algae - would they not use it as shelter and feed on the herbivores? This needs to be more clearly justified. Overall the methods section needs to be reworded in places to ensure the methods applied were relevant and repeatable.

Validity of the findings

The overall study shows that there was a relationship between some taxa and the environmental factors. The modelling is appropriate but does not imply cause and effect, so care is required on some of the discussion material and for some taxa, it might be that they are discussed in the light of all factors influencing them rather than just one at once.

Overall - there are some interesting things in here and the paper should be published, but I would recommend a little more care in the discussion when discussing the relationships of taxa with one environmental factor.

Additional comments

Overall this is an interesting study, and is worthy of publication although I am a little concerned about the replication of the sampling. The authors should be allowed the opportunity to present a revised ms clarifying some of the queries outlined on the main ms.

Reviewer 2 ·

Basic reporting

See author comments

Experimental design

See author comments

Validity of the findings

See author comments

Additional comments

This paper by Momota provides a very thorough investigation on the interactions between the epibiont communities of Zostera marina and the associated motile invertebrates. The study is well planned and constructed and deserves publication in PeerJ, I do however have a number of minor comments that require consideration.

Title: I feel the work is quite novel showing potential cascade impacts of one faunal group on another. Why not make this title bolder that clearly states the key finding of the work.
Abstract: well written, might be useful to have a few quantitative results in their rather than just X influences Y?
Intro: This has a huge amount of references in it. This seems too much.
Results:
Would be nice to see some of the correlations presented visually, especially the interacting ones.
Discussion
Given the importance of the fauna in the present study to fish it would be nice to see more discussion on this. This is of key importance given some of the observed relationships between seagrass density and fish assemblages (e.g. McCloskey et al 2015 PeerJ).
Study has implications for the resilience and functioning of these systems. Would be nice to see a discussion of this and reference to key recent works on resilience and feedbacks in seagrass (e.g. Maxwell et al Biol Rev, Unsworth et al Mar Poll Bull, Eklof et al 2016 J app Ecol).

---

## Round 0.2 · accepted · Accept

You have provided a thorough response to the reviewers comments - thank you. I picked out a few minor edits that can be dealt with at the proofs stage:

229 and 252 fitted rather than fit?
295 amphipods
390 rostrata